# Assessment of Professional Practices in the Care Pathway of Patients with Upper Aerodigestive Tract Cancer in a University Hospital

**DOI:** 10.3390/jcm13216623

**Published:** 2024-11-04

**Authors:** Dounia Chbihi, Morgane Corda, Thomas Thibault, Jérémy Baude, Caroline Guigou, Mireille Folia

**Affiliations:** 1Department of Otolaryngology-Head and Neck Surgery, Croix Rousse Hospital, 69004 Lyon, France; douniachbihi@hotmail.com; 2Department of Otolaryngology-Head and Neck Surgery, Dijon University Hospital, 21000 Dijon, France; morgane.corda@chu-dijon.fr (M.C.); mireille.folia@chu-dijon.fr (M.F.); 3Department of Internal Medicine, Dijon University Hospital, 21000 Dijon, France; thomas.thibault@chu-dijon.fr; 4Department of Radiotherapy, Centre Georges François Leclerc, 21000 Dijon, France; jbaude@cgfl.fr; 5ICMUB Laboratory, UMR CNRS 6302, University of Burgundy, 21000 Dijon, France

**Keywords:** squamous cell carcinoma, upper aerodigestive tract, multidisciplinary team meetings, professional practices

## Abstract

Objectives: The main objective of this study was to evaluate the alignment between treatment decisions made during multidisciplinary team meetings (MTMs) and the treatments received by patients with upper aerodigestive tract cancers. The secondary objective was to identify factors influencing potential discrepancies. Methods: This retrospective, single-center study was conducted at a tertiary referral center and included 147 patients diagnosed with squamous cell carcinoma of the upper aerodigestive tract. Patients were divided into two groups based on the match between MTM-decided and actual treatments. Multivariate analysis was performed to assess factors independently associated with discrepancies. Results: Out of 147 patients, 28 (19%) received treatment that did not align with MTM decisions. Among these, eight died before treatment, one patient refused care, five received supportive care, five patients underwent surgery, three received radiotherapy alone, one patient underwent surgery and adjuvant radiochemotherapy, one patient underwent surgery and adjuvant radiotherapy alone, three patients received radiochemotherapy, and one patient received palliative chemotherapy. Independent significant factors associated with non-concordance included poor performance status (PS) and treatment not received at a tertiary reference center. Treatment shifts mainly involved downgrading from curative to palliative care. Conclusions: This study highlights the importance of patient health status in determining deviations from MTM decisions. Further efforts should focus on improving the integration of patient comorbidities and health status into MTM decision-making to optimize care delivery.

## 1. Introduction

Cancers of the upper aerodigestive tract (UADT) rank as the fifth most diagnosed cancer in France [1]. The various locations, the fragile health conditions of patients, and the range of treatments available warrant a multidisciplinary evaluation [2].

Multidisciplinary team meetings (MTMs) were established and made mandatory in oncology during the 2003–2007 cancer plan by the National Cancer Institute (Institut National du Cancer, INCa) [3]. One of the 70 measures included the implementation of an MTM before any decision-making concerning a cancer patient [3]. The objective of the MTM is to ensure alignment between the decision made and current recommendations, the relevance of the decision considering patient-specific data, and consistency between the proposed and implemented treatment. If discrepancies arise, the referring physician must provide precise arguments [4]. Standardizing decision-making has led to uniform care provision, improved patient follow-up, and better documentation of patient information [5,6,7]. MTMs in oncology have also demonstrated their value in organizing and fostering adherence among practitioners and patients to various treatment plans [5,6,8]. They contribute to better treatment outcomes and shorter waiting times for care [6,9]. MTMs may be under pressure from increasing tumor incidence, a more difficult socio-economic context, physician shortages, and financial pressures [6,10]. In the United Kingdom, to improve MTMs, healthcare professionals use streamlining measures, a process by which complex cases are prioritized for full discussion [10]. The management of simple cases is then accelerated using the standards of care. Nevertheless, the decisions of MTMs can be improved using new technologies such as artificial intelligence [6].

The benefit of MTMs has been demonstrated [6,7,11,12,13], but their practice can still be enhanced. Indeed, the 2018 report from the French National Authority for Health (*Haute Autorité de Santé*, HAS) regarding MTMs in oncology revealed that MTMs were non-compliant with recommendations in 18% of cases. The most commonly identified cause was the presence of fewer than three specialties during the meeting [14].

Few studies report on the alignment between decisions made and treatments actually received by patients with head and neck cancer, especially the factors that may influence changes in treatment plans or their non-execution. However, one study showed better survival when the decisions were followed [9]. Understanding the various causes of discordance could improve decision-making during MTMs, their follow-up, and, consequently, overall patient management. In the absence of scientific articles on this subject, we decided to carry out a retrospective observational study to investigate it.

The main objective of this study was to evaluate the alignment between treatment decisions made during multidisciplinary team meetings (MTMs) and the treatments received by patients with upper aerodigestive tract cancers. The secondary objective was to identify factors influencing potential discrepancies.

## 2. Materials and Methods

### 2.1. Study Design

A retrospective and single-center descriptive study was carried out at a tertiary referral center from January 2019 to January 2020. Data from patients’ medical files were collected.

The inclusion criteria were as follows:-Patients whose files were registered and discussed in MTMs at the University Hospital Centre between 1st January 2019 and 1st January 2020.-Patients diagnosed initially with squamous cell carcinoma of the oral cavity, oropharynx, hypopharynx, and cervical lymphadenopathy without a primary or larynx.-Patients for whom both the decided treatment plan and the executed treatment plan were recorded.

The exclusion criteria were as follows:-Patients with tumor localization in the salivary glands, nasal cavities, or sinuses.-Patients with lesions were categorized as cutaneous tumors.-Patients with histological types other than squamous cell carcinoma.-Patients whose records were presented in MTMs for expert consultation purposes.-Patients with unknown treatment plans.

Patients’ data were anonymized, and data processing was carried out in accordance with the reference methodology MR-004 of the National Commission on Informatics and Liberty (CNIL). This study was performed according to the principles of good clinical practice, and the need for the Ethics Committee approval was waived. Patients were informed and consented to participate in this study. After the examination of this study by Committees for the Protection of Persons (CPP Est I), we found that this trial is outside Jardé’s law field.

### 2.2. Data Collection

The list of patients enrolled in MTMs was provided by the Cancer Coordination Centre of the University Hospital Centre. Various data from the medical file were recorded:-Patients’ characteristics and medical history: age, gender, World Health Organization Performance Status (PS) index, cardiovascular history, diabetes, chronic obstructive pulmonary disease (COPD), cirrhosis, malnutrition history, history of cancer (current or remission), 3-year survival, and alcohol and tobacco use.-Head and Neck Cancer Characteristics: Site of origin (oral cavity, oropharynx, hypopharynx, larynx, and cervical lymphadenopathy without primary), Tumor Node Metastasis (TNM) status at the time of diagnosis presented according to the 8th edition of the UICC (Union for International Cancer Control) 2017 TNM classification for head and neck tumors, histopathology, and human papillomavirus (HPV) status.-Extension Assessment According to the Recommendations of the French Society of Otorhinolaryngology (SFORL) included panendoscopy of the upper aerodigestive tract with the provision of a summary diagram and an operative report, cervical-thoracic CT scan, Ear, Nose, and Throat (ENT) MRI, and Positron Emission Tomography (PET-CT scan).-Multidisciplinary team meeting (MTM): MTMs were held weekly, jointly between the University Hospital Centre and the affiliated cancer center. Each week, it involved at least three physicians from the following specialties: ENT and Maxillofacial Surgery, radiation oncologists, medical oncologists, radiologists, and pathologists. A standardized form was completed and verified by the patient’s referring physician before each meeting, during which the patient’s case was presented, and each aspect of the extension assessment was reviewed. This form included the date, attending physicians, and previous relevant data. Following the meeting, a collective decision on the treatment protocol was made and then validated by the MTM coordinator. The original form was recorded in the Cancer Communicating Folder (CCF). A copy of the form documenting the validated treatment protocol was placed in the patient’s electronic medical record for traceability. Patients were not present at the MTM in the University Hospital Centre. The collective treatment protocol decision was communicated to the patient during the disclosure consultation conducted by the referring ENT physician after the MTM. During this consultation, the patient provided consent or refusal regarding the proposed treatment.

### 2.3. Statistical Analysis

All data tables were collected using Microsoft Excel (version 2022, Redmond, WA, USA). Statistical analyses were performed using R software (version 4.1.2., Miami, FL, USA). Patient characteristics, tumor features, extension assessments, and treatment modalities are expressed as percentages for qualitative variables and as median (interquartile range) for quantitative variables. To compare categorical variables, we relied on bivariate data comparison: Fisher’s exact test was used to compare qualitative variables, and Wilcoxon–Mann–Whitney or student t-test was used to compare quantitative variables according to distribution. The explanatory variables used were (1) patient characteristics, (2) tumor features, and (3) extension assessments to compare the two groups determined by the alignment or non-alignment between the treatment received and the treatment decided in MTMs (MTM—Treatment Matching). Multivariate analysis was performed using logistic regression and the variable “MTM—Treatment Matching” as an outcome to adjust potential confounders. Covariates with a *p*-value less than 0.2 in the bivariate analysis were considered in the multivariate model [15]. A multivariate analysis was performed with a limited number of predictors to avoid overfitting [16]. Multicategorical variables (with more than two categories) were grouped into two categories to meet the log-linearity assumption. The *p*-value < 0.05 was considered as significant.

## 3. Results

### 3.1. Patient Characteristics

The flowchart is detailed in Figure 1. Characteristics of the population are detailed in Table 1. The registration of 343 patients were done and discussed in MTMs. Among them, 148 patients met the exclusion criteria. Out of the 195 treatment decisions, 16 executed treatments were unknown.

We identified 147 patients with an initial diagnosis of squamous cell carcinoma of the head and neck region. The population was predominantly male (sex ratio: 4/1), and the mean age was 64 years (range: 38–93 years). More than half of the patients were in good general condition, with a PS score of 0. Among the main medical histories, we collected cardiovascular histories (ranging from treated hypertension to coronary artery disease) for over half of the patients. A history of cancer other than head and neck cancer was described for more than 16% of the patients. Seventy-five percent of the patients were active or former smokers, and 33% of them had a history of chronic alcohol intoxication.

The TNM stages of the tumors at diagnosis were Tx in 4%, T1 in 16%, T2 in 23%, T3 in 19%, and T4 in 38% of cases. The majority of tumors were diagnosed at an advanced stage. The following locations were identified: oral cavity (39%), larynx (24%), oropharynx (21%), hypopharynx (12%), and cervical lymphadenopathy without primary (4%).

Nodal involvement was present in 50% of cases, and metastatic involvement was observed in 4.7% of cases (seven patients). Tumors were HPV-induced in 15% of cases. We collected extension assessment results for all patients. Panendoscopy of the upper aerodigestive tract was not performed for six patients. Four were contraindicated due to their general condition, and two were refused by the patients. Cervical-thoracic CT scan was refused by two patients. The following additional examinations were conducted: six pulmonary biopsies, two gastrointestinal endoscopies, and two lymph node biopsies.

This extension assessment led to the diagnosis of synchronous cancer in 10% of cases; six synchronous head and neck cancers (two oropharynx, two hypopharynx, one oral cavity, and one larynx), six lung cancers, one esophageal cancer, and one metastatic breast cancer.

### 3.2. Treatment Received

The treatments received or not by the patients are summarized in Table 2. Treatments were administered at the University Hospital Centre and/or the affiliated cancer center in 76.2% of cases. They were curative in over 90% of cases. In total, nine patients did not receive treatment: eight passed away before treatment initiation, and one patient refused treatment.

### 3.3. Treatment Matching

In order to study the various factors that may have influenced the discrepancy between the MTM decision and the treatment received, we stratified the sample into two groups: “Matching” and “Non-matching” (Table 3).

The treatment received by the patient did not match the MTM decision in 28 patients (19%). Of these 28 patients, eight died before treatment, one patient refused care, five received supportive care, five patients underwent surgery, three received radiotherapy alone, one patient underwent surgery and adjuvant radiochemotherapy, one patient underwent surgery and adjuvant radiotherapy alone, three patients received radiochemotherapy, and one patient received palliative chemotherapy. Most commonly, there was a shift from curative treatment to palliative care, or the treatment was downgraded, as shown in Table 3. For one patient, surgical treatment was decided on instead of concomitant radiochemotherapy, as the patient was agoraphobic and could not bear wearing the radiotherapy mask.

The following main factors were significantly associated (Table 4) with the non-matching between the treatment received and the treatment decided in MTM: high PS status (*p* < 0.001), malnutrition at diagnosis (*p* < 0.002), large tumor size (*p* = 0.007), presence of metastases at diagnosis (*p* = 0.003), alcoholism (*p* < 0.001), tobacco (*p* = 0.014), 3-year survival (*p* < 0.001), and the treatment made at tertiary reference center (*p* < 0.001).

There was no significant difference between the “matching” group and the “non-matching” group for medical histories such as COPD (*p* = 0.5), hepatic cirrhosis (*p* = 0.5), or diabetes (*p* > 0.9).

The results of multivariate analysis are shown in Table 5. The variables independently associated with the non-matching between the treatment received and the MTM decision are the high PS status (OR = 1.40, CI95%: 1.12–1.75, *p* = 0.003) and treatment made at tertiary reference (OR = 0.85, CI95%: 0.71–0.94, *p* < 0.007). The other variables, such as large tumor size, presence of metastases, malnutrition, and alcoholism or tobacco use, are no longer associated with the non-matching after multivariate analysis, reflecting confusion bias between these variables and health status measured by the PS.

## 4. Discussion

Our study was conducted within the framework of professional practice evaluation. We found a strong matching between the treatment decisions made in MTM and the treatments administered. The population in our study was representative of patients with UADT, predominantly male and with a history of alcohol and tobacco use.

In a study that recorded protocol changes following weekly MTMs in head and neck cancers, changes were found in one-third of cases (which is a higher proportion than in our study) [17]. The change in treatment protocol was consistent with our study: if a treatment was changed, it was less invasive than the one decided at MTM. MTM evaluation has been conducted in other specialties, such as gynecology in breast cancer [18,19] or hepatogastroenterology [20,21]. Similar alignment rates were found, ranging between 80% and 90%. A large-scale study encompassing brain tumors, head and neck cancers, sarcomas, and musculoskeletal tumors involving over 3000 patients found an MTM treatment matching at 80%, similar to ours [22]. Treatments were predominantly downgraded.

In our study, the factors significantly associated with disagreement between the MTM decision and the treatment received by the patient were poor general health, defined as a PS status of 2 or 3, and treatment not received at a tertiary reference center. In the literature, comorbidities are also found to be factors that predict non-alignment between treatment and MTM decision [20,22]. In the aforementioned studies, these other factors could influence a difference between treatment and the MTM decision. Treatments were redirected toward radiotherapy or chemotherapy protocols [17], again in a trend toward downgrading treatment. Modified treatments secondary to patient refusal or protocol changes by physicians also appeared to be less aggressive than those decided at the MTM [20]. In our study, the observations did not take into account the different physicians. This could be the subject of further studies. The better alignment between the MTM decision and the treatment received when the treatment is carried out in a tertiary referral center may be due to the physicians working there being the ones present at the MTMs and deciding on the treatment.

Some studies suggest that the presence of the referring physician could ensure alignment between the decision made and the treatment received [23]. It could be interesting to study the number of participants and specialties present during MTM, as multidisciplinarity could even be a factor influencing alignment between decisions and treatments administered, according to a study concerning MTMs in neuro-oncology [23].

In our study, when the treatments decided do not correspond to the decisions of the MTM, treatment downgrading is often necessary in clinical practice [24]. This adaptation is often dictated by the patient’s overall health status or the advanced stage of the disease, which may render certain treatments inappropriate. These findings raise questions about the overall evaluation of the health status of patients with head and neck cancers, both in clinical practice and during MTMs [25]. The assessment of comorbidities is thus of major importance in oncological management. Age and malnutrition are significant among the main prognostic factors commonly found in oncology. Although our study did not show a significant influence of age on treatment-MTM matching, it is essential to consider patient age when making therapeutic decisions in MTMs, as it increases the risk of cancer and the number of comorbidities. In our institution, although recommended, geriatric oncology consultation is not yet systematic [26]. Studies have demonstrated its utility, particularly in the diagnoses of pancreatic and colorectal cancers, as well as in therapeutic decisions in onco-hematology [27,28,29]. Malnutrition is also crucial, as a standardized nutritional assessment is part of the recommendations during the diagnosis of head and neck cancers. A thorough analysis of this assessment, in addition to laboratory and dental tests and a geriatric evaluation for patients over 75 years old, allows for a better assessment of the patient’s overall condition and early detection of comorbidities. These comorbidities can lead to diagnostic delays and increased post-treatment mortality. Effective organization of MTMs in oncology is essential to ensure optimal patient management. Studies have shown that structured, weekly, and multidisciplinary MTMs, chaired by a coordinating practitioner, promote alignment between treatment and MTM decisions [30]. Patient presence during MTMs is also sometimes recommended [31], as it allows for a better evaluation of their overall condition and facilitates understanding of their pathology and treatment. Tools such as checklists during MTMs and teamwork evaluation programs have shown their usefulness in improving cohesion and multidisciplinary discussions, contributing to more tailored management of patients with head and neck cancers.

Our work has several limitations. First, this work is a retrospective study. Also, for six patients, curative treatment was proposed even though the patient had died beforehand. It could be important to take better account of the patient’s general condition when deciding on treatment, for example, through anesthesia consultation, a systematic geriatric oncology consultation, or with a form filled in by the ENT surgeon during pan-endoscopy. The patient’s participation in the MTM could help to decide on the appropriate treatment for the patient’s general health, even if this means a longer meeting.

Our study was carried out at a single tertiary referral center. It could have been interesting to carry out a multicenter study to be able to compare the practices of different centers regarding MTM decisions.

This work could be further developed by comparing MTM decisions with the guidelines that will soon be validated by the French Society of Cervico-Facial Carcinology or with decisions made by artificial intelligence. The data collection period may also seem dated (2019 to 2020), as we wanted to include patients treated before the COVID-19 pandemic. We want to further our work, in particular, by comparing the evolution of the alignment between treatment decisions made during MTMs and the treatments received by patients during the COVID-19 pandemic period and immediately afterward.

## 5. Conclusions

Our study was conducted as part of the evaluation of professional practices. It demonstrated that the treatment decisions made during our MTM were mostly aligned with the treatments received. We observed that the patient’s compromised general condition and the treatment made in a tertiary center are independently associated with the non-alignment between treatments administered and decisions made in MTM. This study highlights the importance of a comprehensive oncological assessment and a rigorously conducted MTM. A more comprehensive clinical and biological assessment would prioritize nutritional and supportive care in patient management. Therefore, anticipating precarious conditions should lead to their treatment as soon as the diagnosis is suspected.

## Figures and Tables

**Figure 1 jcm-13-06623-f001:**
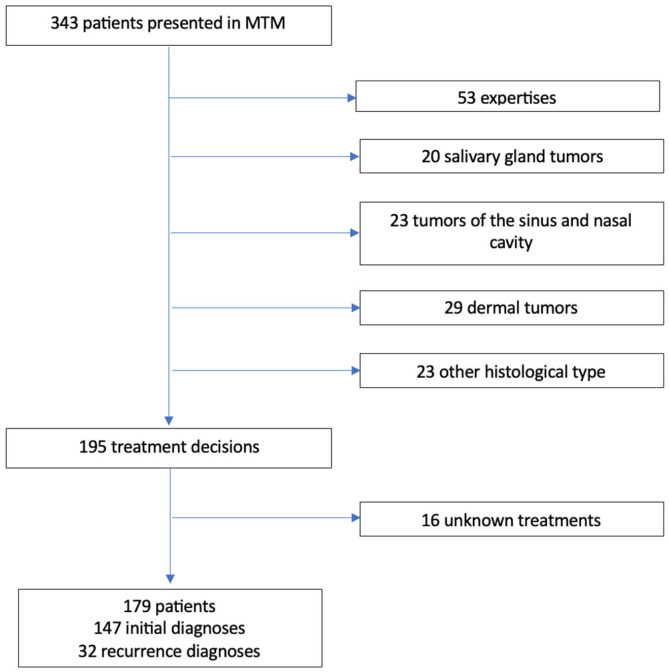
Flowchart. MTM: Multidisciplinary Team Meeting.

**Table 1 jcm-13-06623-t001:** Population characteristics based on medical history and tumor characteristics at the initial diagnosis. sd: standard deviation.

Characteristics	n (%)
Patients	147 (100)
Mean Age (sd)	64 (13)
Sex (men)	117 (80)
Performance Status (PS)	
PS 0	85 (58)
PS 1	50 (34)
PS 2	7 (4.7)
PS 3	5 (3.4)
Medical history	
COPD	39 (26.5)
Cirrhosis	11 (7.5)
Cardiovascular	84 (57.1)
Diabetes	32 (21.8)
Malnutrition	56 (38.1)
Cancer history	
ENT	4 (2.7)
Lung	5 (3.4)
Esophagus	4 (2.7)
Bladder	5 (3.4)
Other	6 (4)
Total	24 (16.3)
Tobacco intoxication	
Mean Pack-year	33
Weaned	59 (40)
Active	21 (14.3)
Chronic alcoholism	
Weaned	21 (14.3)
Active	33 (22.5)
3-year survival	71 (49)

**Table 2 jcm-13-06623-t002:** Patient outcomes or therapeutic modalities following initial diagnosis.

Characteristics	n (%)
Total Patients	147 (100)
Dead before treatment	8 (5.5)
Treatment refused by the patient	1 (0.6)
Supportive care treatment	6 (4.1)
Surgery	30 (20.4)
Radiotherapy alone	20 (13.6)
Surgery and adjuvant radiotherapy	14 (9.5)
Surgery and adjuvant chemoradiotherapy	20 (13.6)
Concomitant chemoradiotherapy	40 (27.2)
Palliative chemotherapy	8 (5.5)

**Table 3 jcm-13-06623-t003:** A cross-tabulation table detailing treatments finally administered or patient outcomes based on MTM decisions in cases of non-matching between treatment and MTM decision at initial diagnosis. MTM = Multidisciplinary Team Meeting. S = Surgery. RT = Radiotherapy. CT = chemotherapy.

	Treatment	S	RT	S + Adjuvant RT	S + Adjuvant CT	Concomitant RTCT	Palliative CT	Supportive Care Treatment	Death Before Treatment	Treatment Refused by the Patient	Total
MTM Decision											
S		0	1	0	0	0	0	0	0	0	1
RT		0	0	0	0	0	0	3	3	0	6
S + adjuvant RT		3	0	0	0	0	0	0	1	0	4
S + adjuvant RTCT		1	0	1	0	2	0	0	0	0	4
Concomitant RTCT		1	2	0	1	0	1	2	2	0	9
Palliative CT		0	0	0	0	1	0	0	2	1	4
Total		5	3	1	1	3	1	5	8	1	28

**Table 4 jcm-13-06623-t004:** Matching between treatment decided in MTM and treatment received at initial diagnosis. MTM = Multidisciplinary team meeting. Welch two sample t-test or Fisher’s exact test.

Variable	MTM—Treatment Matching		*p*-Value
	No (n = 28) n (%)	Yes (n = 119) n (%)	
Patients			
Mean age	67 (57.8)	63 (56.8)	0.3
Sex: Men	25 (89)	92 (77)	0.2
Performance Status (PS)			<0.001
PS 0	6 (21)	79 (66)	
PS 1	14 (50)	36 (30)	
PS 2	6 (21)	1 (0.8)	
PS 3	2 (7.1)	3 (2.5)	
Tobacco (active or weaned)	28 (100)	98 (82)	0.014
COPD	9 (32)	30 (25)	0.5
Alcoholism (active or weaned)	19 (68)	35 (29)	<0.001
Cirrhosis	4 (14)	7 (5.9)	0.2
Cardiovascular history	20 (71)	64 (54)	0.14
Diabetes	2 (7.1)	14 (12)	0.7
Malnutrition	18 (64)	38 (32)	0.002
Cancer history	8 (29)	16 (13)	0.084
Synchronous cancer	4 (14)	10 (8.4)	0.3
3-year survival	4 (14)	67 (56)	<0.001
The treatment made at a tertiary reference center	12 (43)	100 (84)	<0.001
Tumors			
Localization			0.9
Oral Cavity	9 (32)	49 (41)	
Oropharynx	7 (25)	24 (20)	
Hypopharynx	3 (11)	14 (12)	
Larynx	8 (29)	27 (23)	
Cervical lymphadenopathy without primary	1 (3.6)	5 (4.2)	
T stage at diagnosis			0.007
0	1 (3.6)	5 (4.2)	
1	0 (0)	24 (20)	
2	4 (14)	29 (24)	
3	10 (36)	18 (15)	
4	13 (46)	43 (36)	
N stage at diagnosis	19 (68)	69 (58)	0.4
M+ stage at diagnosis	5 (18)	2 (1.7)	0.003

**Table 5 jcm-13-06623-t005:** Multivariate analysis (logistic regression) to predict non-matching between treatment received and MTM decision with variables associated with *p*-value < 0.2 in bivariate analysis. CI = Confidence Interval. OR = Odd Ratio.

Variable	OR	95% CI	*p*-Value
Performance status 2 or 3 (versus 0 or 1)	1.40	1.12, 1.75	0.003
Tobacco (active or weaned)	1.12	0.95, 1.33	0.2
Alcoholism (active or weaned)	1.07	0.93, 1.22	0.4
Cardiovascular history	1.04	0.93, 1.17	0.5
Malnutrition	1.01	0.88, 1.16	0.9
Cancer history	1.09	0.93, 1.27	0.3
The reatment made at the tertiary reference	0.82	0.71, 0.94	0.007
TNM (T3/4 versus T0/1)	1.12	0.99, 1.27	0.065
TNM (N+)	0.98	0.87, 1.10	0.7
TNM (M+)	1.29	0.97, 1.73	0.084

## Data Availability

All article data are available on request by sending an e-mail to the corresponding author.

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
