# Peer review of "Assessment of Professional Practices in the Care Pathway of Patients with Upper Aerodigestive Tract Cancer in a University Hospital"

_jcm, 2024, doi:10.3390/jcm13216623_

Round 1

Reviewer 1 Report

Comments and Suggestions for Authors

The manuscript from Chbihi et al. presents an important study focused on evaluating the alignment of treatment decisions made during Multidisciplinary Team Meetings (MTMs) with the treatments actually received by patients. While the research is of significant clinical interest, several aspects could be refined to enhance the manuscript's impact and clarity. Here are my opinions:

1. Introduction Line 44-59: The introduction could add some of the latest literature to underline ongoing challenges or advancements in MTM practices, particularly in oncology.

2. Can you clarify why the ethics committee approval is waived?

3. It would be beneficial to expand on the methods used for data analysis and any specific software tools beyond R software could improve transparency and reproducibility.

4. Figure 2 needs to be reformed and regenerated by any professional graph software.

5. The discussion may encompass a critical examination of the limitations inherent to the current study, accompanied by a delineation of prospective research directions aimed at enhancing the robustness of the findings.

Comments on the Quality of English Language

 Minor grammatical errors could be corrected to improve readability. A thorough proofreading by a native English speaker is recommended. Such as line 263, 245-248, etc. 

Author Response

Dear reviewer,

All the authors would like to thank you for the time you have spent on our work, and for the quality of your questions and comments.

We hope you will find our answers satisfactory. We remain at your disposal should you feel that any further changes need to be made to the article.

Best regards.

Q1. Introduction Line 44-59: The introduction could add some of the latest literature to underline ongoing challenges or advancements in MTM practices, particularly in oncology.

A1. Thank you for your comment. 3 new references had been added.

Now, this part of introduction had been modified as follow: “MTMs may be under pressure from increasing tumor incidence, a more difficult socio-economic context, physician shortages and financial pressures [6,10]. In the United Kingdom, to improve MTMs, healthcare professionals use streamlining measures, a process by which complex cases are prioritized for full discussion [10]. The management of simple cases is then accelerated using Standards of Care. Nevertheless, the decisions of MTMs can be improved using new technologies such as artificial intelligence [6].” (line 55)

Q2. Can you clarify why the ethics committee approval is waived?

A2. This article was performed in accordance with the principles of good clinical practice. Patients were informed and consented to their participation in this study.

After examination of this study and according to French legislation (because it’s a retrospective study), this trial is outside Jardé’s law field and the need for Ethics Committee approval was waived.

The official document signed by the President of the Committees for the Protection of Persons is attached.

Q3. It would be beneficial to expand on the methods used for data analysis and any specific software tools beyond R software could improve transparency and reproducibility.

A3. Thank you for your comment. The statistics were produced by a medical doctor with specific training in statistics, Mr Thomas Thibault,  who is one of the co-authors.

Now, the paragraph describing the statistical analyses has been rewritten as follows: “All data tables were collected using Microsoft Excel (version 2022, Redmond, Washington, USA). Statistical analyses were performed using R software (version 4.1.2., Miami, Florida, USA). Patient characteristics, tumor features, extension assessments, and treatment modalities are expressed as percentages for qualitative variables and as median (interquartile range) for quantitative variables. To compare categorical variables, we relied on bivariate data comparison: Fisher's exact test was used to compare qualitative variables, and Wilcoxon-Mann-Whitney or student t-test was used to compare quantitative variables according to distribution. The explanatory variables used were: 1) patient characteristics, 2) tumor features, and 3) extension assessments to compare between the 2 groups determined by the alignment or non-alignment between the treatment received and the treatment decided in MTMs (MTM – Treatment Matching). In order to adjust on potentials confounders, multivariate analysis was performed by logistic regression using the variable “MTM – Treatment Matching” as outcome. Covariates with a p-value less than 0.2 in the bivariate analysis were considered in the multivariate model [15]. To avoid overfitting, a multivariate analysis was performed with a limited number of predictors [16]. Multicategorical variables (variables with more than two categories) were grouped into two categories to meet the log-linearity assumption. The p-value <0.05 was considered as significant.” (line 148)

Multivariate analyses were performed, the results section rewritten and a results table added.

Q4. Figure 2 needs to be reformed and regenerated by any professional graph software.

A4. Thank you very much for your comment.

Now, this figure has been removed from the text and the information entered in the manuscript like that: “The TNM stages of the tumors at diagnosis were Tx in 4%, T1 in 16%, T2 in 23%, T3 in 19%, and T4 in 38% of cases.” (line 187).

Q5. The discussion may encompass a critical examination of the limitations inherent to the current study, accompanied by a delineation of prospective research directions aimed at enhancing the robustness of the findings.

A5. Thank you for your comment. The discussion has now been rewritten. Paragraphs have been added to discuss the limits of this work and future work that may follow: “Our work has several limitations. For 6 patients, curative treatment was proposed even though the patient had died beforehand. It could be important to take better account of the patient's general condition when deciding on treatment, for example through the anaesthesia consultation, a systematic geriatric oncology consultation or with a form filled in by the ENT surgeon during pan-endoscopy. The patient's participation in the MTM could help to decide on the appropriate treatment for the patient's general health, even if this means a longer meeting.

Our study was carried out at a single tertiary referral centre. It could have been interesting to carry out a multicentre study to be able to compare the practices of different centres regarding MTMs decisions.

This work could be further developed by comparing MTMs decisions with the guidelines that will soon be validated by the French Society of Cervico-Facial Carcinology, or with decisions made by artificial intelligence. Too, the data collection period may seem dated (2019 to 2020), as we wanted to include patients treated before the COVID-19 pandemic. We would like to further our work, in particular by comparing the evolution of the alignment between treatment decisions made during MTMs and the treatments received by patients during the COVID-19 pandemic period and immediately afterwards.” (line 314)

Q6. Comments on the Quality of English Language

 Minor grammatical errors could be corrected to improve readability. A thorough proofreading by a native English speaker is recommended. Such as line 263, 245-248, etc.

A6. Thank you for your comment. The English has been reworked, especially in the discussion.

Reviewer 2 Report

Comments and Suggestions for Authors

The manuscript is well-written, however some points needs clarification;

-1. I wonder why authors did not exclude the died patients from the study.

- 2. The conclusion in the abstract appear not consistent with the results, kindly revise and update.

-3.  Did not you find any cases of melanoma?

- 4. Factors influencing potential discrepancies should be studied in more details.

Author Response

Dear reviewer,

All the authors would like to thank you for the time you have spent on our work, and for the quality of your questions and comments.

We hope you will find our answers satisfactory. We remain at your disposal should you feel that any further changes need to be made to the article.

Best regards.

Q1. I wonder why authors did not exclude the died patients from the study.

A.1 Thank you for your remark.

These patients were not excluded because a decision to treat them had been made before their death. It is interesting to note that, in some cases, these patients were offered curative treatment (6 patients).

This is a point we need to improve, namely by taking better account of the patient's general condition in the decision.

This was added in the discussion: “Our work has several limitations. For 6 patients, curative treatment was proposed even though the patient had died beforehand. It could be important to take better account of the patient's general condition when deciding on treatment, for example through the anaesthesia consultation, a systematic geriatric oncology consultation or with a form filled in by the ENT surgeon during pan-endoscopy. The patient's participation in the MTM could help to decide on the appropriate treatment for the patient's general health, even if this means a longer meeting.” (line 314)

Q2. The conclusion in the abstract appear not consistent with the results, kindly revise and update.

A2. Now, the conclusion has been modified as follow: “The study highlights the importance of patient health status in determining deviations from MTM decisions. Further efforts should focus on improving the integration of patient comorbidities and health status into MTM decision-making to optimize care delivery. » (line 30)

Q3.  Did not you find any cases of melanoma?

A3. In this work, we have chosen to include only patients with a squamous cell carcinoma. As shown in figure 1, other histological tumor types were presented in MTM but were excluded from the study.

Q4. Factors influencing potential discrepancies should be studied in more details.

A4. Thank you for your remark. The results section has now been rewritten. A multivariate analysis has been performed. A table has been added to show the results.

Reviewer 3 Report

Comments and Suggestions for Authors

In the manuscript, the authors present a study regarding the alignment between treatment decisions made during multidisciplinary team meetings and the treatments received by patients with upper aerodigestive tract cancers. The general idea of the manuscript is quite interesting and nowadays the decisions of the multidisciplinary team in oncology are verry important. The authors included in the study, 195 cases. My observations are :

- In my opinion, the multidisciplinary team must now the general status of the patient. It is not clear in the manuscript why, the most frequent cause of non-compliance with the decisions of the multidisciplinary team is the performance status. It seems that the multidisciplinary team does not new the general condition of the patient. Probably  the referring physician does not take part of the multidisciplinary team meeting. In this case, probably the group of patients would be better to be divided in 2 group. One group in which the referring physician took part at the multidisciplinary team meeting and another group in which the referring physician does not took part of the meeting.

- the statistical analysis is too basic

- the authors stated that in one case the multidisciplinary team decided to perform concomitant RTCT, instead surgery was done. The surgeon was part of the multidisciplinary team ? The surgeon decided on his own decision to perform surgery instead the decision of the multidisciplinary team. Something is missing her.

-please include some limitations of the study at the end of the discussions part of the main manuscript.

Author Response

Dear reviewer,

All the authors would like to thank you for the time you have spent on our work, and for the quality of your questions and comments.

We hope you will find our answers satisfactory. We remain at your disposal should you feel that any further changes need to be made to the article.

Best regards.

Q1. In my opinion, the multidisciplinary team must now the general status of the patient. It is not clear in the manuscript why, the most frequent cause of non-compliance with the decisions of the multidisciplinary team is the performance status. It seems that the multidisciplinary team does not new the general condition of the patient. Probably  the referring physician does not take part of the multidisciplinary team meeting. In this case, probably the group of patients would be better to be divided in 2 group. One group in which the referring physician took part at the multidisciplinary team meeting and another group in which the referring physician does not took part of the meeting.

A.1 Thank you for your remark. The referring doctor was introduced at each MTM.

You're right, the general state of health is not taken into account enough when making decisions about treatment.

This is a critical point in our work, and one that we need to improve. Most patients are only seen once in consultation before pan-endoscopy by the referring physician, who is not always aware of the patient's physiological state.

 A form with the surgeon's opinion during pan-endoscopy, the anaesthetist's medical findings and the generalization of the oncogeriatrician's opinion could be drawn up to take better account of patients' general condition.

This was added to the discussion in this way: “Our work has several limitations. For 6 patients, curative treatment was proposed even though the patient had died beforehand. It could be important to take better account of the patient's general condition when deciding on treatment, for example through the anaesthesia consultation, a systematic geriatric oncology consultation or with a form filled in by the ENT surgeon during pan-endoscopy. The patient's participation in the MTM could help to decide on the appropriate treatment for the patient's general health, even if this means a longer meeting.” (line 314)

Q.2 the statistical analysis is too basic

A.2 Thank you for your comment. The statistics were produced by a medical doctor with specific training in statistics, Mr Thomas Thibault, who is one of the co-authors. A multivariate analysis has now been performed. The results section has been rewritten in the same way as the section describing the statistical analyses.

Q.3 The authors stated that in one case the multidisciplinary team decided to perform concomitant RTCT, instead surgery was done. The surgeon was part of the multidisciplinary team ? The surgeon decided on his own decision to perform surgery instead the decision of the multidisciplinary team. Something is missing her.

A.3 Thank you for your comment.

The referring surgeon was present at every MTMs.

Concerning this patient, he was agoraphobic and couldn't stand the radiotherapy mask. The request for surgical treatment came from the patient.

This information has now been added to the text as follows: “For one patient, surgical treatment was decided on instead of concomitant radiochemotherapy, as the patient was agoraphobic and could not bear wearing the radiotherapy mask.” (line 227)

Q.4 please include some limitations of the study at the end of the discussions part of the main manuscript.

A.4 Now, this paragraph has been added: “Our work has several limitations. For 6 patients, curative treatment was proposed even though the patient had died beforehand. It could be important to take better account of the patient's general condition when deciding on treatment, for example through the anaesthesia consultation, a systematic geriatric oncology consultation or with a form filled in by the ENT surgeon during pan-endoscopy. The patient's participation in the MTM could help to decide on the appropriate treatment for the patient's general health, even if this means a longer meeting. Our study was carried out at a single tertiary referral centre. It could have been interesting to carry out a multicentre study to be able to compare the practices of different centres regarding MTMs decisions.This work could be further developed by comparing MTMs decisions with the guidelines that will soon be validated by the French Society of Cervico-Facial Carcinology, or with decisions made by artificial intelligence. Too, the data collection period may seem dated (2019 to 2020), as we wanted to include patients treated before the COVID-19 pandemic. We would like to further our work, in particular by comparing the evolution of the alignment between treatment decisions made during MTMs and the treatments received by patients during the COVID-19 pandemic period and immediately afterwards.” (line 314)

Round 2

Reviewer 3 Report

Comments and Suggestions for Authors

In the manuscript, the authors present a study regarding the efficiency of the oncologic multidisciplinary team in patients with upper aero-digestive tract cancer. The manuscript has been reviewed before and the authors changed the manuscript according to the previous reviewer indications. Their comments are pertinent. The quality of the manuscript has been improved. I have also another suggestion : please include the fact that the study is retrospective in the limitations of the study. 

Author Response

Q.1 In the manuscript, the authors present a study regarding the efficiency of the oncologic multidisciplinary team in patients with upper aero-digestive tract cancer. The manuscript has been reviewed before and the authors changed the manuscript according to the previous reviewer indications. Their comments are pertinent. The quality of the manuscript has been improved. I have also another suggestion : please include the fact that the study is retrospective in the limitations of the study.

A.1 Thank you for your comment. Now this sentence has been added in the discussion: “Our work has several limitations. First, this work is a retrospective study. » (line 317)